# Validating Consensus-Defined Severity Grading of Lymphatic Complications after Kidney Transplant

**DOI:** 10.3390/jcm10214858

**Published:** 2021-10-22

**Authors:** Mohammadsadegh Sabagh, Sara Mohammadi, Ali Ramouz, Elias Khajeh, Omid Ghamarnejad, Christian Morath, Markus Mieth, Yakup Kulu, Martin Zeier, Arianeb Mehrabi, Mohammad Golriz

**Affiliations:** 1Department of General, Visceral and Transplantation Surgery, University of Heidelberg, 69120 Heidelberg, Germany; Mohammadsadegh.Sabagh@med.uni-heidelberg.de (M.S.); Sara.mohamadi.kh@gmail.com (S.M.); Ali.Ramouz@med.uni-heidelberg.de (A.R.); Elias.Khajeh@med.uni-heidelberg.de (E.K.); Omid.ghd@gmail.com (O.G.); Markus.Mieth@med.uni-heidelberg.de (M.M.); Yakup.Kulu@med.uni-heidelberg.de (Y.K.); Mohammad.Golriz@med.uni-heidelberg.de (M.G.); 2Department of Nephrology, Heidelberg University Hospital, 69120 Heidelberg, Germany; Christian.Morath@med.uni-heidelberg.de (C.M.); Martin.Zeier@med.uni-heidelberg.de (M.Z.)

**Keywords:** kidney transplantation, lymphatic complications, lymphocele, severity grading, validation

## Abstract

Lymphatic complications after kidney transplantation (KTx) are associated with morbidities such as impaired wound healing, thrombosis, and organ failure. Recently, a consensus regarding the definition and severity grading of lymphoceles has been suggested. The aim of the present study was to validate this classification method. All adult patients who underwent KTx between December 2011 and September 2016 in our department were evaluated regarding lymphoceles that were diagnosed within 6 months after KTx based on the recent definition. Patients with lymphoceles were categorized according to the classification criteria, and clinical outcomes were compared between the groups. In our department, a total of 587 patients underwent KTx between 2011 and 2016. Lymphoceles were detected after KTx in 90 patients (15.3%). Among these patients, 24 (26.6%) had grade A lymphoceles, 14 (15.6%) had grade B, and 52 (57.8%) had grade C. The median duration times of intermediate care (IMC) and hospital stay were significantly higher among patients with grade C lymphoceles than they were among patients with grade A and B lymphoceles. Significantly more patients with grade C lymphoceles were readmitted to the hospital for treatment. The recently published definition and severity grading of lymphoceles after KTx is an easy-to-use and valid classification system, which may facilitate the comparison of results from different studies on lymphoceles after KTx.

## 1. Introduction

Owing to surgical developments, kidney transplantation (KTx) has become a routine treatment for patients with end-stage kidney disease. Although the morbidity and mortality rates after KTx are acceptable [1], the rate of lymphocele formation after KTx is still high. Lymphoceles can lead to further complications, such as impaired wound healing, thrombosis, and organ failure. The reported incidences of lymphoceles after KTx vary widely (from 0.6% to 51%) [2,3,4]. This vast discrepancy in the reported incidences can be attributed to the lack of a standardized definition for post-KTx lymphatic complications but also to differences in the means of routine postoperative diagnosis and distinct follow-up protocols [5,6]. Furthermore, the variation in definitions regarding lymphatic complications after KTx restricts the ability to carry out the meaningful comparison of results between studies [7]. An accepted definition of lymphatic complications after KTx is needed to allow for the comparison of results between studies and to help clinicians make evidence-based decisions regarding surgical practice. 

Mehrabi et al. addressed the lack of an objective definition for lymphoceles that follow KTx [4] in a systematic review that evaluated existing definitions and suggested a consensus-defined severity grading for post-KTx lymphoceles. They found that lymphoceles were defined as a fluid collection of any size near the transplanted kidney after urinoma, hematoma, and abscess were excluded. Based on their review, Mehrabi et al. classified lymphoceles as grade A, B, or C depending on how severely they affected clinical management [4]. However, to date, this classification has not been validated in an independent patient cohort.

The aim of this study was to verify the proposed definition and severity grading in a retrospective cohort analysis of KTx recipients in our visceral transplantation center.

## 2. Materials and Methods

This study included all patients who underwent kidney transplantation (KTx) between December 2011 and September 2016 in our department. All lymphoceles diagnosed within six months of KTx were included in this study. 

### 2.1. Surgical Team and Surgical Procedure

KTxs were performed by transplant surgeons in our department. All organs were prepared during the back-table procedure, and extra tissues near the hilum of the kidney and around the ureter were resected after ligation on the kidney side. Implantations were performed using a standardized KTx procedure via an extraperitoneal approach, making a J-shaped incision in the abdominal wall (the “hockey stick”). The peritoneum was mobilized from the psoas muscle, the arterial and venous iliac axes were dissected, the lymphatic vessels were ligated, and the renal artery and vein were anastomosed via an end-to-side technique to the iliac artery (common or external) and the external iliac vein, respectively. Using an extravesical ureteroneocystostomy approach, the ureter was anastomosed to the urinary bladder [8]. All patients were fitted with a single-J stent, a transurethral and a suprapubic catheter, and two easy-flow drains. 

### 2.2. Data Extraction

#### 2.2.1. Donor-Related Data

The baseline and demographic characteristics of the graft donors, including the donor status (living or brain dead), gender, age, side of the donated graft, and cold ischemia time, were collected. 

#### 2.2.2. Preoperative Data

The characteristics of the recipients, such as age, gender, body mass index (BMI), smoking status, comorbidities (chronic anemia, coagulopathy, hypertension, diabetes mellitus), underlying renal diseases, type and duration of dialysis (months), and previous abdominal operation (nephrectomy or retransplantation), were collected.

#### 2.2.3. Intra- and Postoperative Data 

Intraoperatively, the side of the graft implantation (left/right), duration of operation (minutes), and blood loss (mL) were recorded. The main immunosuppressive regimens (cyclosporin or tacrolimus, based on our previously published study [9]), laboratory data, and imaging reports (ultrasound, CT, and MRI) were recorded. All patients underwent routine postoperative ultrasound check-ups every day before discharge and at every outpatient follow-up. The easy-flow drains were removed between days 3 and 7 after the fluid secretion decreased and there were no signs of bleeding, lymphorrhoea, or leakage (urinoma). The duration times of the intermediate care unit (IMC) and hospital stay after KTx and any subsequent lymphocele-related admissions were also recorded. 

Lymphoceles were diagnosed and classified according to the definition and grading system proposed by Mehrabi et al. (Table 1). Lymphoceles were considered to be loculated if one or more septa were present inside the fluid collection. Lymphocele-related complications were classified into general, visceral, vascular, and urological complications. During follow-up, lymphocele diagnosis, management approach, and mortality were evaluated. In addition, recurrence of lymphoceles was also documented, taking into account the primary grade of lymphocele according to the first treatment approach.

### 2.3. Statistical Analysis

This was an explorative study, and the sample size could not be calculated. IBM SPSS Statistics for Windows version 22.0 (IBM Corp. Released 2013. Armonk, NY, USA) was used for statistical analyses. Categorical data are presented as proportions and percentages, and the empirical distribution of continuous data is presented as the mean ± standard deviation or as the median (range). The analysis was performed using the analysis of variances (ANOVA) test for continuous data and chi-squared or Fisher’s exact test for categorical data to define the factors associated with clinically relevant lymphoceles following a kidney transplant. A *p*-value < 0.05 was considered to be statistically significant. To define the correlation between lymphocele severity and postoperative complications (categorical data), a post hoc analysis was carried out by calculating the standardized residuals of crosstabulation ((frequency-expected)/standard error). The two-sided α level was adjusted to 0.008 (= 0.05/(2×3)) using Bonferroni correction for multiple comparisons. Due to the fact that there was a significant difference after the ANOVA, an appropriate post hoc test (the Bonferroni test) was applied to compare the subgroups.

## 3. Results

### 3.1. Pre- and Intraoperative Data

During the study period, 587 patients underwent KTx at our department (Figure 1). The mean recipient age was 51.4 ± 13.6 years. In total, 39.8% of the patients were female. Glomerulonephritis (42.0%) was the most common reason for KTx, followed by congenital renal disease (27.4%), and the majority of patients had a history of preoperative dialysis (93.0%), with a mean duration of 63.7 (1–274) months. The most common comorbidity was hypertension (75.2%). Diabetes mellitus was observed in 14.6% of recipients. KTx was performed on the right side in 50.3% of the patients and on the left side in 49.7% of patients.

Lymphoceles were detected in 90 patients (15.3%). The mean age of recipients diagnosed with lymphoceles was 54.4 ± 12.0 years, and 70% of these patients were male. Obesity (BMI > 30) was reported in 24% of these patients. The most common reason for KTx in the patients diagnosed with lymphoceles was glomerulonephritis (39%), and 92% of these patients had a median (range) history of preoperative dialysis of 65 (2–186) months. Common comorbidities included chronic anemia and hypertension (73%) and diabetes mellitus (12%). Kidney transplantation was performed on the right side in 47 patients (52.2%) and on the left side in 43 patients (47.8%).

### 3.2. Postoperative Data

As shown in Figure 1, 90 patients were diagnosed with lymphoceles after KTx, with the highest rate being 57.8% for grade C (*n* = 52). Of these 52 patients, 40 patients underwent surgical intervention after non-surgical interventions were unsuccessful. All grade A lymphoceles were asymptomatic and did not require intervention. General symptoms were reported in 12.2% of KTx patients with lymphoceles (infection in four patients, wound dehiscence in five patients, and fever in two patients). Visceral symptoms were detected in 34.4% of KTx patients with lymphoceles (abdominal pain in 21 patients, abdominal swelling in 4 patients, and mass in 6 patients). Vascular symptoms were documented in 26.6% of KTx patients with lymphoceles (vein compression in 7 patients, deep vein thrombosis in 4 patients, and leg edema in 13 patients). Urological symptoms were reported in 20% of KTx patients with lymphoceles. The mean (range) duration of hospital stays following KTx was 31 (7–182) days. No mortality was documented during the first 90 days. One-year survival was 98.9%, and the mortality that did occur was not associated with complications regarding lymphoceles. 

### 3.3. Validation of Lymphocele Severity after KTx

The characteristics of patients who had grade A, B, and C lymphoceles are shown in Table 2. No differences were detected in pre- or intraoperative factors, including common comorbidities, cold ischemia time, side of graft implantation, duration of operation, and blood loss between the groups.

Data regarding the diagnosis, symptoms, and management of post-KTx lymphoceles are presented in Table 3. The prevalence of visceral and vascular manifestations was significantly higher in patients with grade C lymphoceles (0% in grade A, 57.1% in grade B, and 75% in grade C, *p* < 0.001). Significantly more urological symptoms were observed in patients with grade C lymphoceles (*p* < 0.001). Plasma creatinine levels recorded at the time of lymphocele diagnosis were not significantly different in the three grades of lymphoceles (grade A: 2.2 (1.0–6.1), grade B: 1.9 (0.8–4.4), grade C: 1.6 (0.8–6.3), *p* = 0.883). However, an increase in creatinine level and/or graft dysfunction were significantly more common in patients with grade C lymphoceles (none in grade A, 3 in grade B, and 15 in grade C, *p* = 0.014). Although the post hoc analysis showed no correlation between graft dysfunction and grade B/C lymphoceles (post hoc *p* = 0.317 and 0.012, respectively), grade A lymphoceles were significantly associated with a lower rate of graft dysfunction (post hoc *p* = 0.003). 

The loculation of lymphoceles was not reported in 42 of the 90 patients (46.6%). Additionally, an uniloculated lymphocele was detected in 17 patients (18.8%), and 25 patients (27.7%) developed a multiloculated lymphocele. Albeit not significant, the rate of multiloculated lymphoceles was higher in patients with grade C lymphoceles.

The primary management of post-KTx lymphoceles took a non-surgical approach in 78 patients (86%) and a surgical approach in 12 patients (14%). Lymphoceles were persistent or recurrent in 44 patients. Failed primary non-surgical intervention was documented for 40 patients (44.4%). There was no lymphocele recurrence in patients with grade A lymphoceles, whereas recurrence was detected in 43 of 54 patients (79.6%) who underwent primary non-surgical intervention and 2 of 12 patients (16.6%) who underwent a primary surgical approach. Among patients with primary grade B lymphoceles, 40 patients were treated surgically after lymphocele recurrence and were subsequently graded as having grade C lymphoceles. In this study, the rate of lymphocele recurrence after the non-surgical intervention was higher than that after surgical intervention (*p* < 0.001). Post hoc analysis revealed that grade A lymphoceles were reciprocally associated with recurrence (*p* < 0.001), whereas primary grade B lymphoceles had the highest recurrence rate, which was significant (*p* < 0.001). 

The readmission rate following the first approach was highest among patients with primary grade B lymphoceles (27.8%), followed by patients with grade C lymphoceles (8.3%). None of the patients with grade A lymphoceles were readmitted to the hospital following KTx. The readmission rate was significantly different among the three grades of lymphoceles (*p* = 0.008). The post hoc analysis showed a significantly higher readmission rate in patients with grade B lymphoceles (*p* = 0.002).

The duration of IMC stay was significantly different among the three grades of lymphoceles (grade A: 4 (0–5) days, grade B: 4 (2–8) days, grade C: 5 (2–11) days; *p* < 0.001), and the median duration of IMC stay was significantly higher for patients with grade C lymphoceles than for patients with grade A and B lymphoceles (*p* < 0.001 and *p* = 0.027, respectively). Grade C lymphoceles were associated with a longer hospital stay duration, compared with grade A and B lymphoceles (*p* < 0.001 and *p* = 0.041, respectively).

## 4. Discussion

Lymphatic formations are common complications following KTx, which affect the patients’ quality of life but can also lead to rehospitalization, reoperation, and even secondary graft loss in some cases [10]. The reported incidence of lymphoceles following KTx varies between studies [2,4]. This wide range of reported incidences between studies may be explained through the investigation of different postoperative outcomes and by the use of different definitions for lymphoceles. These inconsistencies in patient selection and management strategies between studies mean that analytic comparisons cannot be made [5,11,12,13,14,15,16]. Furthermore, different conservative, non-surgical, and surgical methods have been used to treat lymphoceles [6,7,17], and these treatments cannot be compared because of the heterogeneity of the data. To address this conflict, a consensus regarding the definition and severity grading of lymphoceles following KTx was developed for both asymptomatic and symptomatic lymphoceles [4]. This classification categorized post-KTx lymphocele formation into grades A, B, or C based on the ultimate clinical intervention needed (Table 1). Here, we validated this classification in an independent patient cohort.

In our study, the incidence of lymphoceles after KTx was 15.3%, which was within previously reported ranges (0.6–51%). This included symptomatic and asymptomatic lymphoceles. The most common lymphocele grade was grade C, followed by grade A, and then grade B. The management and postoperative care of lymphoceles after KTx determined the lymphocele grade [2,4]. The rates of visceral, vascular, and urological symptoms were highest among patients with grade C lymphoceles, followed by patients with grade B lymphoceles, in accordance with the definition. 

The risk factors identified for lymphocele formation after KTx can be categorized into donor-related and recipient-related factors [2]. Donor-related factors are mostly related to organ preparation; for instance, preparing organs from a deceased donor rather than a living donor with the precise ligation of lymph nodes in the hilum may decrease the chance of lymphocele formation [18]. This shows that organs from living donors are more likely to develop lymphoceles following KTx. Moreover, it has been suggested that a longer warm ischemia time (caused by longer dissection) significantly increases the chance of lymphocele formation following KTx [19]. Although not statistically significant, a higher rate of grade B and C lymphoceles in the recipients of living donor organs was also seen in the current study.

The definition proposed by Mehrabi et al. does not discuss the etiology of lymphoceles. The grading helps one to create a realistic comparison between the different reported rates of postoperative lymphatic formation following kidney transplantation. In comparison with the Clavien–Dindo classification [20,21], this classification is specific to lymphocele formation following kidney transplantation and can more effectively categorize the complications for use in a statistical comparison. It states that lymphoceles requiring surgical intervention are classified as grade C. In addition, lymphoceles that are surgically managed during an operation for other reasons or those that develop due to iatrogenic complications of grade A/B lymphoceles (e.g., bleeding, intestinal perforation) are also classified as grade C [4]. There is considerable disagreement in the literature regarding the management of lymphoceles [2,22]. Several studies have described surgery as the first-line treatment for lymphocele, according to the center policy and management approach [13,17,23]. However, in our study, primary surgical treatment was only indicated as preferable in 12 patients (2.04%). Most of the grade C lymphoceles (6.8%) were operated on because the previous non-surgical intervention was not successful or because lymphoceles had recurred. Thus, failed non-surgical treatment approaches likely explain the high rate of grade C lymphoceles in our cohort, in agreement with the classification proposed by Mehrabi et al. This finding is consistent with earlier reports in the literature that emphasized the efficacy of surgical treatment in preventing recurrence [15,24]. Accordingly, patients with surgically managed lymphoceles were readmitted less frequently than individuals with grade B lymphoceles. However, the surgical management of grade C lymphoceles increased the durations of IMC and hospital stay, which was due to the necessity of pre- and postoperative care for these patients. Postoperative hospitalization is longer in our center because of the established procedures that are followed. After a KTx, patients are directly transferred to an IMC for transplanted patients. Thereafter, patients are monitored in normal surgical wards, and finally, they are transferred to the Department of Nephrology, where discharge occurs. This process elongates the entire postoperative hospitalization period in comparison to that of other countries. 

Finally, our analysis demonstrated that depending on the postoperative course, the grading of lymphoceles facilitates a valid classification system which enables a reliable comparison between the different reported rates of postoperative lymphatic complications and the severity of their impact on the clinical course of the patient. 

## 5. Conclusions

In conclusion, the grading system proposed by Mehrabi et al. seems to be a valid method for use in classifying lymphoceles following KTx. However, further validation through the use of prospective studies is recommended.

## Figures and Tables

**Figure 1 jcm-10-04858-f001:**
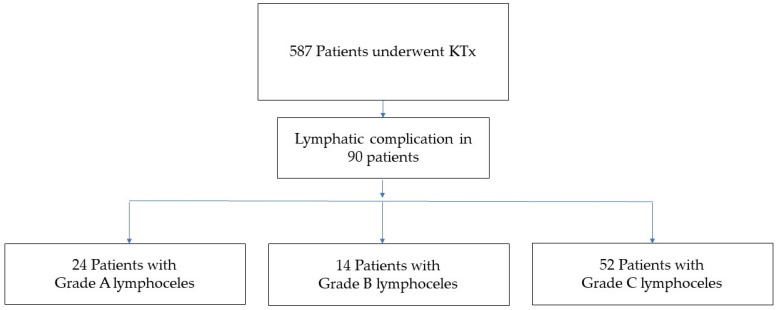
Diagram showing the total number of kidney transplantations and number of patients with postoperative lymphatic complications, divided into Grade A, B, and C.

**Table 1 jcm-10-04858-t001:** Severity grading of lymphatic complications after kidney transplantation based on the consensus from Mehrabi et al. [4].

	Grade A	Grade B	Grade C
Treatment	None/diagnostic or therapeutic aspiration (puncture)	Non-surgical intervention	Surgical treatment (laparoscopic/open)
		(percutaneous external drainage, sclerotherapy, double-J, radiation)	
Persistence	No	No/yes	Yes
Loculation	Non-loculated	Non-loculated	Non-loculated/loculated
Infection	No	No/yes	No/yes
Rise of serum creatinine levels	No	Usually no	No/yes
Previous failed non-surgical intervention	No	No/yes	Usually yes
Previous failed surgical intervention	No	No	No/yes

**Table 2 jcm-10-04858-t002:** Donors’ and recipients’ perioperative data categorized based on the grade of lymphocele formation following kidney transplantation with comparison between the groups.

	Grade A *n* = 24 (4.1%)	Grade B *n* = 14 (2.4%)	Grade C *n* = 52 (8.8%)	*p*-Value
**Donors**				
Brain dead/living donor	16/8 (66.7/33.3)	4/10 (28.6/71.4)	25/27 (48.1/51.9)	0.070
Gender (female/male)	12/12 (50/50)	8/6 (57.1/42.9)	30/22 (57.7/42.3)	0.814
Age (year)	54.9 ± 11.4	59.2 ± 12.9	55.0 ± 11.8	0.342
Side of the kidney (left/right)	11/13 (45.8/54.2)	7/7 (50/50)	27/25 (51.9/48.1)	0.885
**Recipients**				
Gender (female/male)	10/14 (41.6/58.4)	5/9 (35.7/64.3)	12/40 (23.1/78.9)	0.228
Age (year)	51.3 ± 13.0	55.1 ± 12.8	51.2 ± 14.9	0.556
BMI (kg/m^2^)	26.1 ± 3.8	27.1 ± 6.2	26.3 ± 4.0	0.859
Smoking	9 (37.5)	4 (28.6)	22 (42.3)	0.637
**Indication for KTx**				0.581
Glomerulonephritis	10 (41.7)	5 (35.7)	21 (41.4)	
Congenital/polycystic disorder	9 (37.5)	5 (35.7)	15 (28.8)	
Diabetes/hypertension	1 (4.1)	1 (7.1)	6 (11.5)	
Obstructive nephropathy	1 (4.1)	1 (7.1)	1 (1.9)	
Peripheral vascular disease	0 (0.0)	0 (0.0)	4 (7.7)	
Tubulointerstitial nephritis	1 (4.8)	2 (14.3)	1 (1.9)	
Other/unknown	2 (8.2)	0 (0.0)	4 (7.7)	
**Dialysis before KTx**				0.245
None	0 (0.0)	2 (14.3)	5 (9.3)	
Hemodialysis	22 (90.5)	9 (64.3)	37 (71.1)	
Peritoneal dialysis	2 (9.5)	3 (21.4)	10 (19.2)	
Duration	73.8 (9.0–171)	81.7 (3–186)	57.8 (2–147)	0.196
**Comorbidities**				
Chronic anemia	14 (58.3)	13 (92.9)	40 (71.4)	0.069
Coagulopathy	0 (0.0)	0 (0.0)	3 (5.4)	0.322
Hypertension	14 (58.3)	13 (92.9)	40 (71.4)	0.051
Diabetes mellitus	2 (9.5)	2 (14.3)	7 (12.5)	0.791
**Previous abdominal operation**	5 (23.8)	7 (50)	18 (34.6)	0.176
Nephrectomy	3 (14.3)	3 (21.4)	10 (19.2)	0.719
Retransplantation	0 (0.0)	2 (14.3)	7 (12.5)	0.162
Cold ischemia time (hours)	10.4 (0.0–20.0)	10.9 (3.3–14.0)	11.3 (0.0–33.1)	0.950
Side of graft (left/right)	13/11 (54.2/45.8)	9/5 (64.3/35.7)	21/31 (40.4/59.6)	0.216
Duration of operation (minutes)	178.8 (105–332)	169.5 (110–265)	188.8 (90–480)	0.803
Blood loss (mL)	357.5 (100–1700)	234.6 (50–500)	394.3 (50–2200)	0.632
Main immunosuppression (cyclosporin/tacrolimus)	14/10 (58.3/41.7)	6/8 (42.9/57.1)	34/18 (65.4/34.6)	0.306
Creatinine level (at diagnosis, mg/dL)	2.2 (0.6–6.0)	2.4 (0.9–5.3)	2.2 (0.8–6.2)	0.856

Abbreviations: BMI: body mass index; KTx: kidney transplantation.

**Table 3 jcm-10-04858-t003:** Different manifestations of lymphocele formation following kidney transplantation with related postinterventional results, divided and compared based on the grade of lymphocele following kidney transplantation.

	Grade A *n* = 24 (4.1%)	Grade B *n* = 14 (2.4%)	Grade C *n* = 52 (8.8%)	*p*-Value ^§^ (*pv* < 0.05)	Lymphocele Grade	Post Hoc * (*pv* < 0.008)
**Symptomatic lymphocele**	0 (0.0)	14 (100)	52 (100)	**<0.001**	**Grade A**	**<0.0001**
					**Grade B**	0.01
					**Grade C**	**<0.0001**
General manifestations	0 (0.0)	2 (14.3)	9 (17.3)	0.098		
Visceral manifestations	0 (0.0)	7 (50)	24 (46.2)	<0.001		
Vascular manifestations	0 (0.0)	5 (35.7)	19 (36.5)	0.003		
Urological	0 (0.0)	3 (21.4)	15 (28.8)	**0.014**	**Grade A**	**0.003**
					**Grade B**	0.317
					**Grade C**	0.012
**Loculation**				0.660		
Uniloculated	5 (23.8)	2 (14.3)	10 (19.2)			
Multiloculated	4 (16.7)	4 (28.6)	17 (32.7)			
**Outcome**						
Recurrence after 1st intervention *	0/24 (0.0)	43/54 (79.6)	2/12 (16.6)	**<0.001**	**Grade A**	**<0.001**
					**Grade B**	**<0.001**
					**Grade C**	0.193
Readmission after 1st intervention *	0/24 (0)	15/54 (27.8)	1/12 (8.3)	**0.008**	**Grade A**	**0.006**
					**Grade B**	**0.002**
					**Grade C**	0.368
IMC stay (days)	4 (0–5)	4 (2–8)	5 (2–11)	**<0.001**	**Grade A vs. B**	0.157
					**Grade B vs. C**	**<0.001**
					**Grade A vs. C**	**0.027**
Hospital stay (days)	21.22 (10–46)	26.1 (9–91)	30.9 (12–67)	**<0.001**	**Grade A vs. B**	**0.011**
					**Grade B vs. C**	**0.041**
					**Grade A vs. C**	**<0.001**

^§^ This table compares the pre- and postoperative data between patients with different grades of lymphocele. Categorical and continuous data were analyzed using chi-squared test and ANOVA, respectively. * Post hoc analysis was carried out in case of significant outcomes. Abbreviations: IMC: intermediate care unit; KTx: kidney transplantation.

## Data Availability

By rational request from contributing author, the data can be provided.

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
