# Peer review of "Validating Consensus-Defined Severity Grading of Lymphatic Complications after Kidney Transplant"

_jcm, 2021, doi:10.3390/jcm10214858_

Round 1

Reviewer 1 Report

This is a very interesting paper on lymphoceles after kidney transplantation.

The authors report on a cohort of 587 kidney transplant patients, of whom 90 developed a lymphocele.

86% of the lymphoceles were managed non-surgically.

The authors then validated the lymphocele classification by Mehrabi et al.

The manuscipt is very detailed, as are the tables.

I have the following comments:

  • the authors might consider adding table captions
  • I do not quite understand the statistics in Table 3 - some short explantations at the bottom of the table would be good
  • the discussion is quite long and may be shortened

Reviewer 2 Report

This paper examines the validity and effectiveness of the classification of post-kidney transplant lymphocele proposed by Mehrabi et al.

However, the following addition should be required.

  1.  Whether the kidney transplant procedure, especially the kidney graft procedure, is ligating the tissue around the kidney or the tissue around the ureter? Or, do the authors use LigaSure in place of ligation. Similarly, recipient surgery should describe in detail how the lymphatic vessels around the arteries are treated.
  2. .Grading depends on the success of conservative therapy and is unlikely to be suitable for future study of the etiology of lymphocele.
  3. Evidence that this classification is clinically valid should be provided.
  4. Recently, it has been pointed out that mTOR inhibitors (Certican) may cause lymphocele, so immunosuppressive methods for each group should be added. 

Round 2

Reviewer 2 Report

The author takes the requests from the reviewers seriously, responds to them and corrects them. Although it has little scientific basis, it has some clinical usefulness.